# Preparation of a Novel Lignin Nanosphere Adsorbent for Enhancing Adsorption of Lead

**DOI:** 10.3390/molecules24152704

**Published:** 2019-07-25

**Authors:** Chao Liu, Youming Li, Yi Hou

**Affiliations:** 1State Key Laboratory of Pulp and Paper Engineering, South China University of Technology, Guangzhou 510640, China; 2National Engineering Research Center of Papermaking and Pollution Control, South China University of Technology, Guangzhou 510640, China

**Keywords:** lignin, carboxymethyl, nanosphere, adsorption, lead

## Abstract

Carboxymethyl lignin nanospheres (CLNPs) were synthesized by a two-step method using microwave irradiation and antisolvent. The morphology and structure of CLNPs were characterized by ^31^P-NMR, FTIR, and SEM, and the results showed that they had an average diameter of 73.9 nm, a surface area of 8.63 m^2^ or 3.2 times larger than the original lignin, and abundant carboxyl functional groups of 1.8 mmol/g. The influence of dosage, pH, contact time, and concentration on the adsorption of metal ions onto CLNPs were analyzed, and the maximum adsorption capacity of CLNPs for Pb(II) was found to be 333.26 mg/g, which is significantly higher than other lignin-based adsorbents and conventional adsorbents. Adsorption kinetics and isotherms indicated that the adsorption of lead ions in water onto CLNPs followed the pseudo-second-order model based on monolayer chemisorption mechanism. The main chemical interaction between CLNPs and lead ions was chelation. CLNPs also showed an excellent recycling performance, with only 27.0% adsorption capacity loss after 10 consecutive adsorption–desorption cycles.

Academic Editors: Francesco Tornabene and Rossana Dimitri

## 1. Introduction

Heavy metal ions in water can be transmitted and enriched in the food chain in the ecosystem, posing a threat to the surrounding environment and to human health [1]. Toxic heavy metals, such as lead (Pb), copper (Cu), cadmium (Cd), nickel (Ni), chromium (Cr), and zinc (Zn), have been classified as major pollutants, which cause numerous diseases and disorders at even very low levels. Therefore, toxic heavy metal ions must be removed from wastewater effluent before being discharged into the environment. The removal of heavy metals by electrochemical treatment, adsorption, membrane technology, and other effective methods has been extensively studied [2,3,4]. In recent years, various highly efficient materials, such as carbon nanotubes, activated carbon, graphene, molecular sieves, and polymer materials, have been used as adsorbents for wastewater purification and been widely commercialized [5,6,7]. However, due to its high cost, poor adsorption capacity, and poor regeneration performance, its industrial application is limited [6,8]. These limitations have prompted a search for low-cost and efficient adsorbents as replacements. 

Lignin is the second most abundant natural polymer in lignocellulosic biomass (15–30 wt %) next to cellulose [9]. At least 50 million tons of industrial lignin is produced annually from the pulp and paper industry, but less than 2% of the lignin is separated from black liquor and transformed into other high-value products [10,11]. In fact, the combination of abundance, biodegradability, low cost, and abundant active groups of lignin makes it a promising raw material for the preparation of adsorbents [12,13]. However, the adsorption capacity of lignin adsorbents for metal ions is low, such as 9.0 mg/g for Pb(II), 7.5 mg/g for Cd(II), 26 mg/g for Cu(II), and 17.97 mg/g for Cr(III) [6,14], which limits their practical application in the treatment of water that is contaminated with heavy metals.

At present, the method for increasing the adsorption of lignin is mainly lignin modification or increasing the specific surface area of lignin. Dizhbite et al. [15] studied the introduction of oxygen-containing groups by oxidative modification of organic solvent lignin from wheat straw. After modification, the COOH content of lignin increased significantly. At pH 5.0 and 20 °C, the saturated adsorption capacity of the modified lignin Pb(II) reached 155.4 mg/g. Peternele et al. [16] reported a carboxymethyl-formate-modified lignin with a saturated adsorption capacity of 107.5 mg/g for Pb(II). Li et al. [17] prepared a lignosulfonate-based porous lignin microsphere, but its adsorption capacity for heavy metal lead was only 27.1 mg/g. The lignin microspheres prepared by Ge et al. [8] had a diameter of 348 μm, a specific surface area of 9.6 m ^2^/g or 5.3 times that of the original lignin, and an adsorption capacity of 33.9 mg/g for Pb(II). There is an urgent need to develop a new method for the synthesis of lignin that can increase the specific surface area of lignin based on the completion of lignin modification. In our previous work, ionic liquids (ILs) were used as “green solvents” for the formation of alkali lignin nanospheres (ALNPs). The method involved the preparation of lignin nanospheres at high concentration, which could effectively improve the yield of lignin nanospheres [18]. Compared with traditional adsorbents, nanospheres have larger surface area and better diffusion, dispersion, and mass transfer behavior. Meanwhile, many studies have shown that, in addition to phenolic hydroxyl groups, alcoholic hydroxyl groups can also be reacted with sodium monochloroacetate under microwave irradiation for alkali lignin (AL) [19,20]. However, to date, there have been no reports on carboxyl-modified lignin nanospheres.

In this study, we report on a novel lignin nanosphere with abundant carboxyl functional groups, which has the advantages of a simple preparation method, strong adsorption capacity for lead ions, and recyclability. The lignin nanospheres, which were prepared using ionic liquid and antisolvent, greatly reduced the water solubility of carboxyl-modified lignin and improved the chelating ability of carboxyl with heavy metal ions (Figure 1). We also studied the carboxyl content and chemical structure of the modified lignin. The adsorption of lead(II) on carboxymethyl lignin nanospheres (CLNPs) was studied by controlling the pH, reaction time, adsorbent dosage, and initial concentration. The adsorption mechanism was studied by analyzing adsorption thermodynamic, isotherms, and kinetic parameters. Moreover, the performance of the adsorbent was compared with that of some existing adsorbents.

## 2. Experimental

### 2.1. Materials

1-methylimidazole (>99 wt %, Acros, Guangzhou, China), bromoethane (>99 wt %, Aladdin, Guangzhou, China) and ethyl acetate (>99.9 wt%, Aladdin, Guangzhou, China) were used to prepare ionic liquids [EMIM][Ac] in-house according to a procedure available in the literature [21]. Lignin (alkali lignin, ≥99% purity) and sodium monochloroacetate were purchased from Sigma-Aldrich (Beijing, China). All chemicals used were of analytical reagent grade.

### 2.2. Preparation of CLNPs

CLNPs were prepared by a two-step method that involved synthesizing carboxyl-modified AL and preparing carboxymethyl lignin nanospheres. For the first step, 1.5 g alkali lignin, 1.17 g sodium monochloroacetate, and 0.4 g sodium hydroxide were dissolved in 30 mL distilled water under stirring. After that, the mixtures were reacted in a microwave device (XH-100B, XINGHU Microwave Beijing, China) and then microwave-treated for 30 min at 95 °C. The pH of the solution was adjusted to neutral with dilute sulfuric acid. It was then filtered and washed by deionized water and dried under vacuum at 50 °C for 24 h. For the second step, 1 g carboxymethyl lignin was dissolved in 10 mL [Emim][Ac] using a microwave catalytic synthesis extractor. The dissolving conditions were as follows: microwave power, 500 W; temperature, 80 °C; reaction time, 30 min. Then, the solution was stirred at room temperature (25 °C) with magnetic stirring speed of about 600 rpm. After these steps, acid water (pH = 2–3) was gradually added into the solution at the speed of 3 mL/min to 80 vol %, and stirring was continued for 60 min to form CLNPs. After particle formation and solvent removal, the particles were centrifuged at 9000 rpm for 10 min and washed twice with deionized water. Finally, CLNP was freeze-dried and stored in a sealed vial. 

### 2.3. Characterizations

To measure the carboxyl group content of the lignin samples, ^31^P-NMR was carried out using Bruker AVANCE 600 NMR spectrometer (Bruker, Karlsruhe, Germany) according to existing research methods [22]. FTIR, SEM, and the size of CLNPs were tested in our previous study [18]. The Brunauer–Emmett–Teller (BET) method was utilized to calculate the surface area [23]. X-ray photoelectron spectroscopy (XPS) was applied on a polycrystalline X-ray diffraction instrument (D8 ADVANCE, Karlsruhe, Germany). Binding energy values were calibrated using characteristic carbon (C1s = 284.8 eV) during data processing of XPS spectra.

### 2.4. Adsorption

Pb(II) adsorption onto CLNPs was measured by mixing CLNPs with aqueous solutions. All experiments were conducted in a 150 mL conical flask containing 50 mL of a Pb(II) solution that was placed in a constant temperature oscillator at 30 ± 2 °C. Then, the supernatant was removed by filtration. The filtrate was analyzed by a Z-2000 atomic absorption spectrophotometer. All adsorption experiments were performed three times, and the average was calculated to avoid any experimental error. The removal efficiency (*E*) and adsorption amount (*Q_e_*) were calculated by the following equations:(1)$$E(\%)=\frac{C_{0}-C_{e}}{C_{0}}\text{×}100$$
(2)$$Q_{e}\;(\mathrm{mg}/g)=\frac{C_{0}-C_{e}}{m}\times V$$
where *C*_0_ and *C_e_* are the initial and final concentration of Pb(II) (mg/L), respectively; V is the volume of Pb(II) solution (ml); and *m* is the mass of CLNPs (g). After adsorption of the metal ions, the lead-loaded CLNPs were desorbed in HNO_3_ (0.1 M) solution for 4 h at 25 °C. Then, it was neutralized by NaOH (0.1 M), filtered and washed with deionized water until the pH remained constant, and then lyophilized. The regenerated CLNP sample was used for a further adsorption–desorption test of 10 cycles to investigate its recyclability. 

## 3. Results and Discussion

### 3.1. Characterizations

NMR analysis was performed to detect the carboxyl group content of the functional group. The corresponding NMR signal of AL and CLNPs is shown in Figure 2. The corresponding quantitative results of the functional group are shown in Table 1. The carboxyl contents of AL and CLNPs were 0.95 and 1.80 mmol/g, respectively. This effectively increased the carboxyl content of the product, which was 0.90 times higher than that of AL. It can also be seen from Table 1 that sodium monochloroacetate mainly reacted with the phenolic hydroxyl group in the alkali lignin during microwave irradiation, which is consistent with reports in the related literature [20]. From the FTIR spectra (Figure 3), the strength of 1710 cm^−1^ was attributed to the carboxyl group. Compared with AL and ALNPs, CLNPs had more obvious chemical information of carboxyl functional groups. The bands at 1510 cm^−1^ represented the aromatic skeletal vibrations, which indicated that the aromatic structure of lignin was not damaged during the synthesis of carboxyl-modified lignin nanospheres. The FTIR analysis agreed well with the results of the NMR analysis. This also proves that the microwave-assisted synthesis of carboxyl-modified lignin nanospheres can not only improve the reaction efficiency in a short time but also retain the original benzene ring structure of the alkali lignin. 

The morphology of CLNPs was observed by SEM. As shown in Figure 4, the CLNPs were found to be uniformly spherical. The average particle size of nanospheres, as determined by dynamic laser light scatterometer, was approximately 73.9 nm (Figure 5A). The surface area of CLNPs was determined by the BET method from the N^2^ adsorption isotherm, as shown in Figure 5B. The S_BET_ of CLNPs was 8.63 m^2^/g, which was 3.2 times that of lignin (2.72 m^2^/g). The large surface area is beneficial for the adsorption of lead in wastewater.

Previous studies have suggested that lignin has a strong self-assembly capability [24]. According to our group’s previous research, they coexist in the form of single molecules and aggregate in solution. After the antisolvent is added to the solution, the hydrophobic chains of the complexes continue to aggregate to form a stable nanoparticle core [18]. This process should result in a bimodal distribution of particle size unless it reaches the equilibrium, which requires a very long time. Interestingly, the ionic liquid molecules with a weaker hydrophobicity are still dispersed in solution [18]. With improved antisolvent, nanospheres composed of a hydrophilic shell and a hydrophobic core are formed. The self-assembly method largely reduces the water solubility of the nanospheres, so more carboxyl group-containing functional groups are exposed on the surface of the nanospheres. 

### 3.2. Adsorption Studies

The pH of the solution affects the presence and solubility of heavy metal ions in water. The pH affects the surface charge and the extent of functional stratification of the adsorbent material, and the pH of the solution plays an important role in the adsorption of metal ions. The removal efficiency of heavy metal ions by CLNPs and LNPs was examined within the pH efficiency of 2.04–7.06, and the results are depicted in Figure 6. As can be seen, the adsorption amount of CLNPs was significantly higher than that of LNPs, and the adsorption capacity increased until the pH increased to 6.03. When the pH was low, too many hydrogen ions in the solution protonated the carboxyl group of CLNPs to form –COOH, which weakened the electrostatic attraction and complexing ability between the carboxyl group and the heavy metal ion, thereby reducing the adsorption amount. The carboxyl group on the CLNPs deprotonated further with the increase in pH value, and COO^−^ was combined with heavy metal ions to increase the adsorption amount. At pH = 6.03, the adsorption capacity of CLNPs (217.21 mg/g) was 4.2 times higher than LNPs (51.67 mg/g). Furthermore, it is worth noting that when the pH exceeds the pH threshold (>6) of Pb(OH)_2_ precipitation, the removal process is a combination of precipitation of Pb(OH)_2_ and adsorption [25]. Therefore, to avoid the formation of metal hydroxide precipitation, pH = 6.03 was chosen for further adsorption test.

The effect of CLNPs dosage on adsorption metal ions at pH = 6.03 was also studied. The test was carried out under conditions of an initial lead ion concentration of 100 mg/L for 180 min. As shown in Figure 7, the adsorption capacity of CLNPs initially increased with CLNPs loading and reached 333.26 mg/g at 0.3 g/L. As the adsorption dosage was further increased, the adsorption capacity began to decrease. Initially, as the amount of CLNPs increased, the increase in adsorption capacity was due to an increase in the effective adsorption sites on the surface of the adsorbent. In contrast, as the initial amount of lead ions was kept constant, the adsorption amount of Pb(II) decreased with an increase in the adsorbent dosage. This might have been due to an increase in the CLNPs dose, which resulted in a more unsaturated adsorption site on the surface of the adsorbent, thereby resulting in a decrease in adsorption capacity.

Adsorption kinetics were studied (*C_0_* = 100 mg/L, 30 °C, CLNP dosage of 15 mg/50 mL, pH = 6.03) to determine the equilibrium time and adsorption rate. As illustrated in Figure 8, the results showed that the adsorption of metal ions increased sharply at the beginning and reached saturation within 180 min. The initial rapid adsorption was due to the availability of the initial large number of vacancies and the chelating force for mass transfer. Subsequently, the filling of the vacancies became difficult owing to the repulsive force between the Pb(II) adsorbed on the surface of the nanosphere and the Pb(II) in the bulk solution [8]. As shown in Table 2, the saturated adsorption capacity of CLNPs was 333.26 mg/g, which is much higher than other lignin-based adsorbents and conventional adsorbents, such as those shown in Table 2. The high adsorption capacity might have been due to its large surface area and large amount of accessible carboxyl groups.

The kinetic data were also analyzed by the pseudo-first-order (Equation (3)) [26] and pseudo-second-order kinetic models (Equation (4)) [27]:(3)$$\log\left(Q_{e}-Q_{t}\right)=\log Q_{e}-\frac{K_{1}}{2.303}t$$
(4)$$\frac{t}{Q_{t}}=\frac{1}{K_{2}Q_{e}^{2}}+\frac{t}{Q_{e}}$$
where *Q_t_* and *Q_e_* are the amounts of metal ions adsorbed (mg/g) at contact time *t* (min) and at equilibrium, respectively; *k_1_* (1 min^−1^) and *k_2_* (g/mg min) are the rate constants. The kinetic process of Pb(II) in water by CLNPs is shown in Figure 8 and Table 3. It can be observed that the *Q_e_* calculated by the pseudo-first-order model did not match the experimental adsorption amount, and the *R^2^* value was very low (0.6013). This indicates that the first-order model cannot describe the adsorption kinetics. For the pseudo-second-order model, the calculated *Q_e_* (350.9 mg/g) was close to the experimental value, and a higher correlation coefficient (*R^2^* = 0.9991) was obtained. This suggests that the adsorption followed pseudo-second-order kinetics. The pseudo-second-order is based on the chemical adsorption between the metal ion and the active sites of the adsorbent [25]. The results show that the adsorption of Pb(II) onto CLNPs was controlled by chemical adsorption involving chelating, electrostatic forces ion exchange, and valence forces between the adsorbent and adsorbate.

Langmuir [32] and Freundlich [33] isotherm models were applied to fitting Pb(II) adsorption data. The model can be represented by Equations (5) and (6):(5)$$\mathrm{Langmuir}:\;\frac{C_{e}}{Q_{e}}=\frac{C_{e}}{Q_{max}}+\frac{1}{Q_{max}b}$$
(6)$$\mathrm{Freundlich}:\;\log Q_{e}=\log K_{F}+\frac{1}{n}\log C_{e}$$
where *C_e_* (mg/L) is the equilibrium concentration, *Q_e_* (mg/g) is the equilibrium adsorption capacity, *b* (L/mg) is the Langmuir constant, *Q _max_* (mg/g) is the maximum adsorption capacity, and *K _F_* (mg/g) and *n* are the Freundlich constants. The adsorption isotherms at different initial concentration in the range of 20–160 mg/L were collected. It can be seen from Figure 9 and Table 4 that the adsorption amounts of Pb(II) by CLNPs increased significantly with the increase in Pb(II) concentrations until a stable level was reached. The sharp increase in adsorption capacity was observed at low concentrations due to excessive active sites and strong electrostatic attraction, chelating forces for mass transfer. Compared to the Freundlich model (*R^2^* = 0.448), the Langmuir model provided better fitting results (*R^2^* = 0.973). The experimental data agreed well with the Langmuir adsorption experiment, indicating that lead ions were adsorbed on the surface of CLNPs by a monolayer pattern.

In order to clarify the adsorption mechanism of Pb(II) onto CLNPs, XPS analysis of metal-loaded CLNP was performed. Figure 10 shows the XPS spectra of CLNPs before and after adsorption of heavy metals. In addition to the characteristic peaks of C1s (283.82 eV) and O1s (530.95 eV) in CLNPs, the characteristic peak of Pb4f was also detected in the XPS full-spectrum trace of CLNPs after adsorption of heavy metals, which was 139.40 eV. This indicated that heavy metal ions had been successfully adsorbed onto the surface of CLNPs. The O1s peaks of CLNPs that had adsorbed heavy metal were fitted (Figure 10). In the spectrum of O1s, characteristic peaks of oxygen-containing groups (COO^−^, C=O, and C–OH) appeared at 531.34 and 530.15 eV. A comparison of the spectra of Ols before and after adsorption of heavy metals by CLNPs showed that the binding energies of the two characteristic peaks of O1s were increased (532.2 and 530.57 eV, respectively), which indicated that the oxygen atoms on the surface of CLNPs formed a coordination structure with Pb(II). It was formed by the chelation of COO^−^, C=O, and C–OH groups with Pb(II), as shown in Figure 11. This indicated that CLNPs adsorbed heavy metal ions in water by chemical adsorption, which is in agreement with the analysis of kinetic fitting.

To investigate regeneration and the reusability of CLNPs, five adsorption–desorption cycles were performed on Pb(II). Regeneration of the lead-loaded CLNPs can be realized by desorbing with HNO_3_ solution (0.1 M), neutralizing with NaOH (0.1 M), and then washing. As displayed in Figure 12, CLNPs had a good stability for adsorption of Pb(II). After five consecutive adsorption–desorption cycles, there was only a 15.0% loss in adsorption capacity. After 10 consecutive adsorption–desorption cycles, the adsorption capacity loss was 27.0%. This has great practical guiding value for the recycling performance of materials. The test results show that CLNPs can be used as an environmentally friendly and low-cost water purification adsorbent and that Pb(II) has good recyclability on CLNP, which is beneficial for practical applications.

## 4. Conclusions

In this work, eco-friendly and recyclable lignin nanospheres for lead removal were synthesized by the regulation of carboxyl-functionalized alkali lignin and ionic liquid interface. The CLNP adsorbent with higher carboxyl content (1.8 mmol/g) and surface area (8.63 m^2^) had an average diameter of 73.9 nm, which is significantly superior to the traditional raw lignin.

The CLNPs showed a high lead adsorption capacity (333.26 mg/g) and removal efficiency (98.5%). The adsorption kinetics of lead followed the pseudo-second-order model, and the adsorption process was close to the chemisorption process through monolayer adsorption. Their main adsorption mechanism was chelation between CLNPs and lead ions. Moreover, the adsorbed Pb(II) metal ion could be easily desorbed from CLNP adsorbent using 0.1 M HNO_3_ solution. The CLNPs had good reusability for adsorption of metal ions, which means it has great potential for industrial applications in wastewater treatments. However, wastewater has complex components of inorganic salts, organic matters, and heavy metals ions, which is far different from the ideal or real wastewater. CLNPs therefore still have significant room for improvement in terms of practical applications.

## Figures and Tables

**Figure 1 molecules-24-02704-f001:**
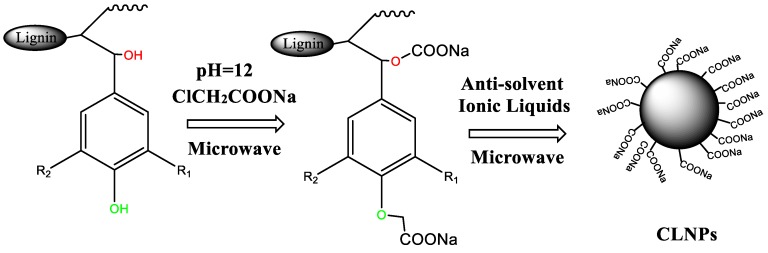
Schematic diagram showing the two-step synthesis of carboxymethyl lignin nanospheres (CLNPs).

**Figure 2 molecules-24-02704-f002:**
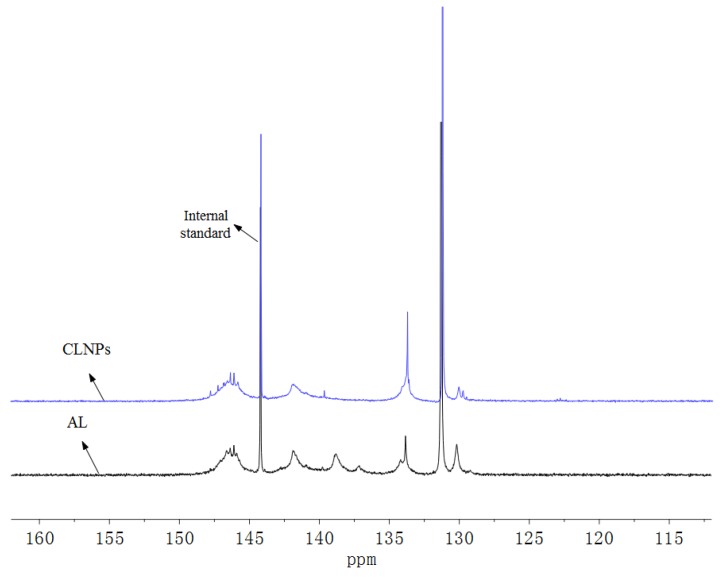
Quantitative ^31^P-NMR spectra of the alkali lignin (AL) and CLNPs.

**Figure 3 molecules-24-02704-f003:**
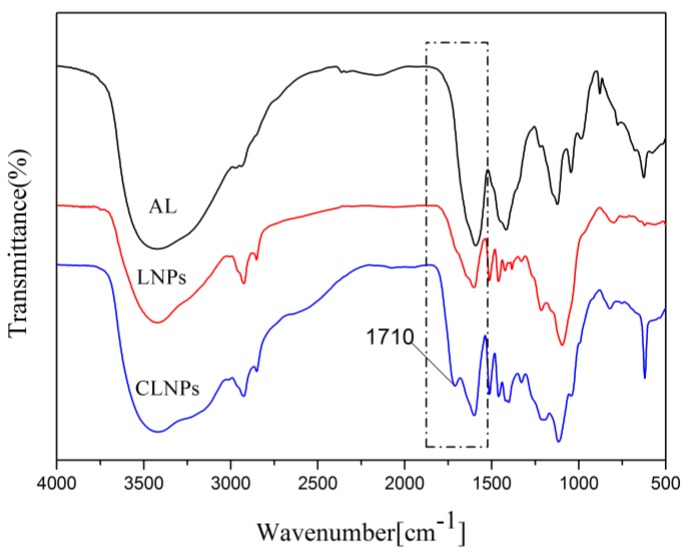
IR spectra of AL, lignin nanospheres (LNPs), and CLNPs.

**Figure 4 molecules-24-02704-f004:**
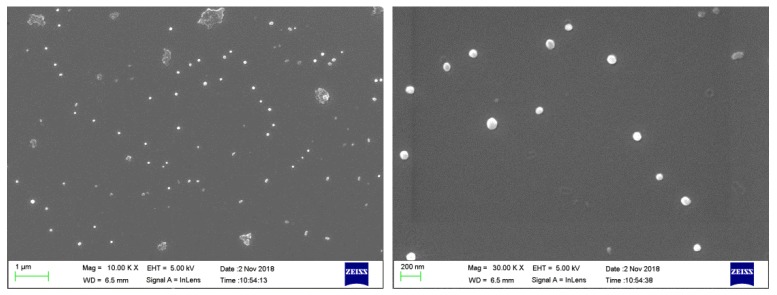
SEM images of CLNPs.

**Figure 5 molecules-24-02704-f005:**
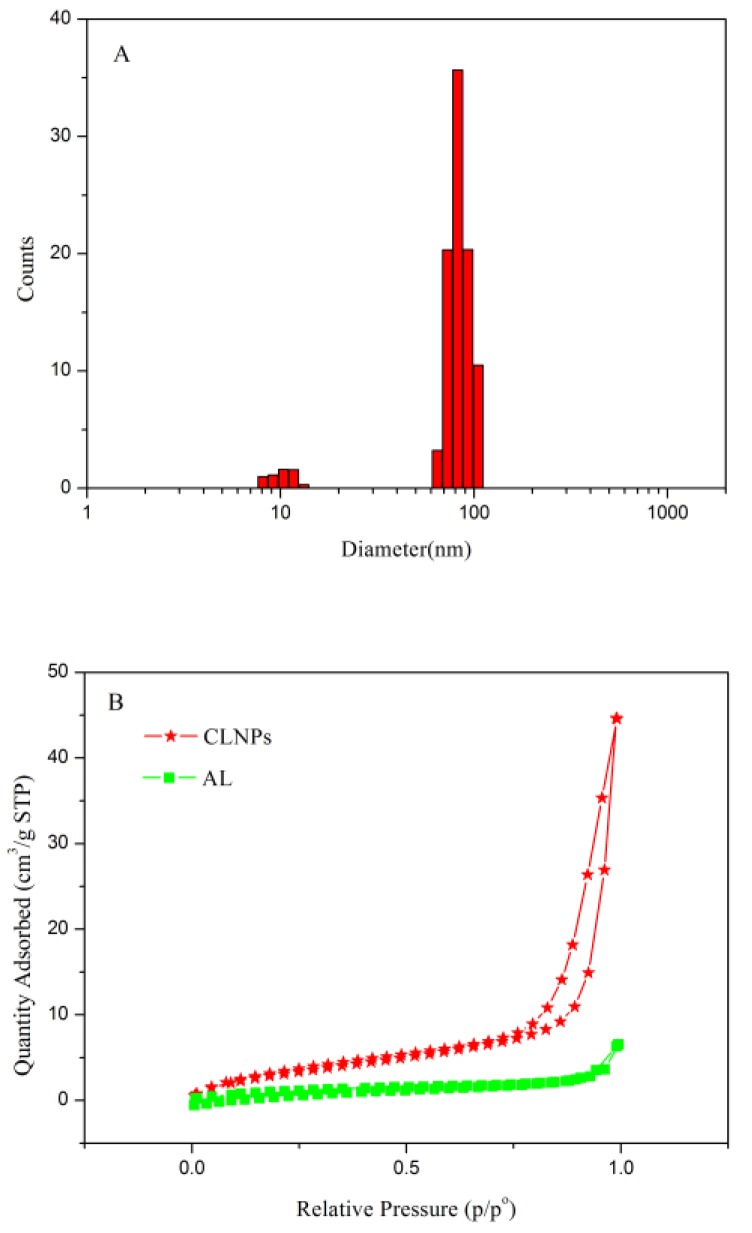
Particle size distribution of CLNPs (**A**) and nitrogen adsorption–desorption isotherms (**B**).

**Figure 6 molecules-24-02704-f006:**
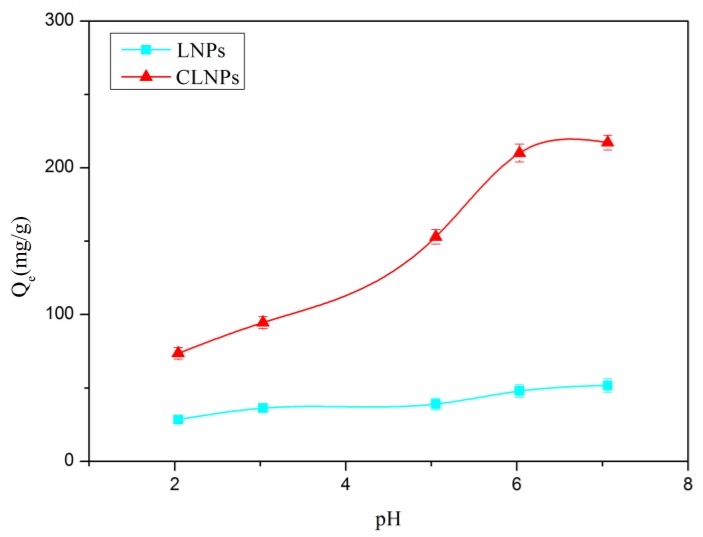
Effect of pH on the adsorption amount of Pb(II) by CLNPs and LNPs (dosage = 10 mg/50 mL, *C_0_* = 100 mg/L, t = 180 min, temperature = 30 °C).

**Figure 7 molecules-24-02704-f007:**
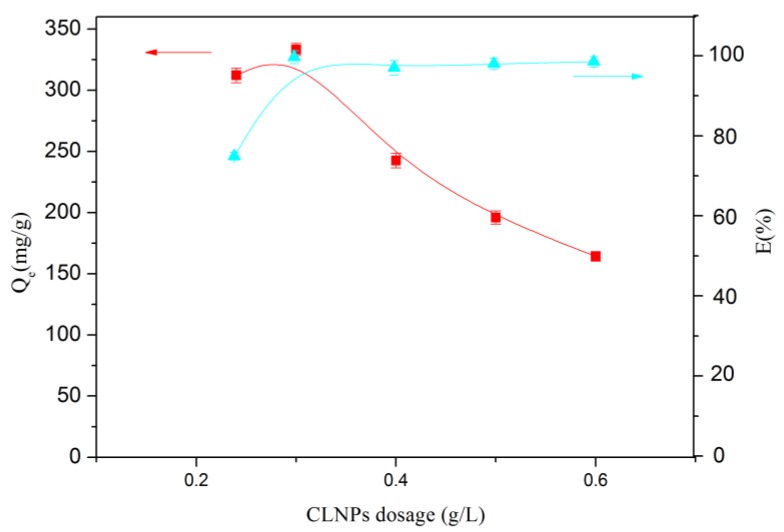
Effect of dosage on the adsorption of Pb(II) on CLNPs (*C*_0_ = 100 mg/L, t = 180 min, temperature = 30 °C, pH = 6.03).

**Figure 8 molecules-24-02704-f008:**
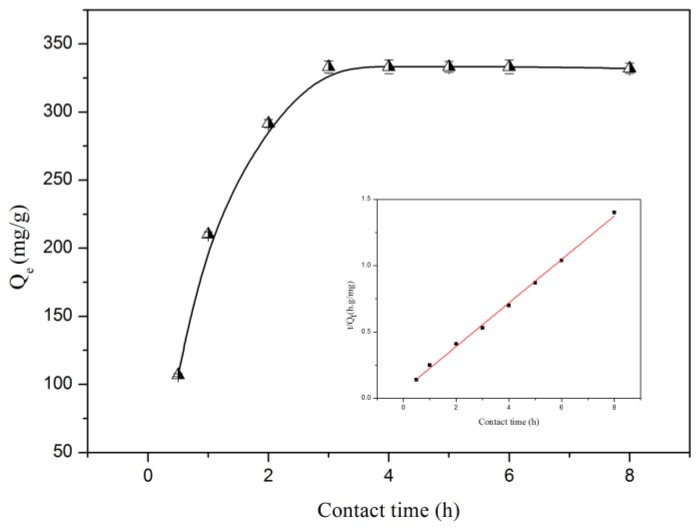
Kinetic adsorption results of Pb(II) on CLNPs. The inset shows the fitting results of the pseudo-second-order model for Pb(II) adsorption (CLNP dosage = 15 mg/50 mL, *C*_0_ = 100 mg/L, temperature = 30 °C, pH = 6.03).

**Figure 9 molecules-24-02704-f009:**
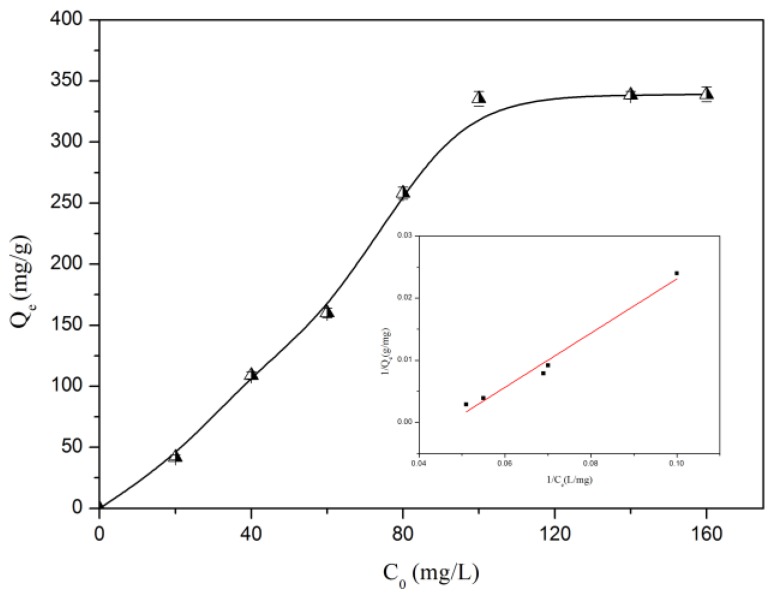
Isotherms of Pb(II) adsorption on CLNPs. The inset shows the fitting results of the Langmuir model for Pb(II) adsorption (CLNP dosage = 15 mg/50 mL, temperature = 30 °C, pH = 6.03).

**Figure 10 molecules-24-02704-f010:**
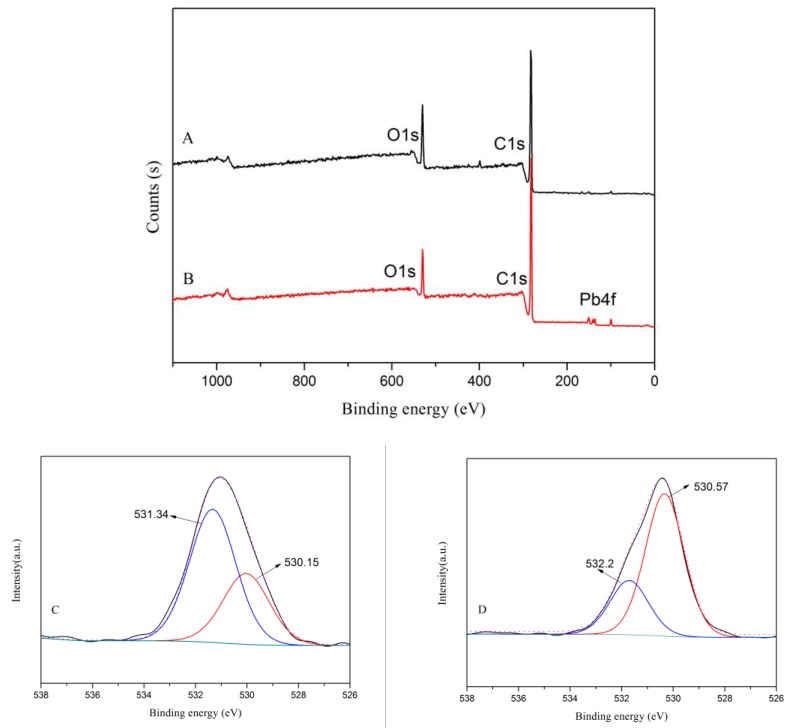
The typical XPS wide scan spectra of CLNPs before (**A**) and after (**B**) Pb(II) adsorption. O1s spectra of CLNPs before (**C**) and after (**D**) Pb(II) adsorption.

**Figure 11 molecules-24-02704-f011:**
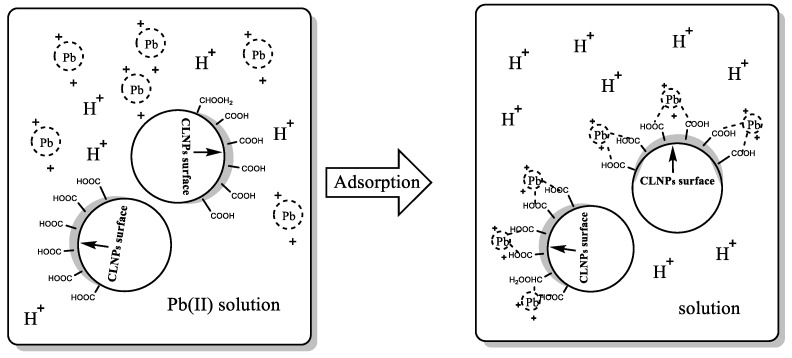
A potential mechanism of Pb(II) adsorption on CLNPs.

**Figure 12 molecules-24-02704-f012:**
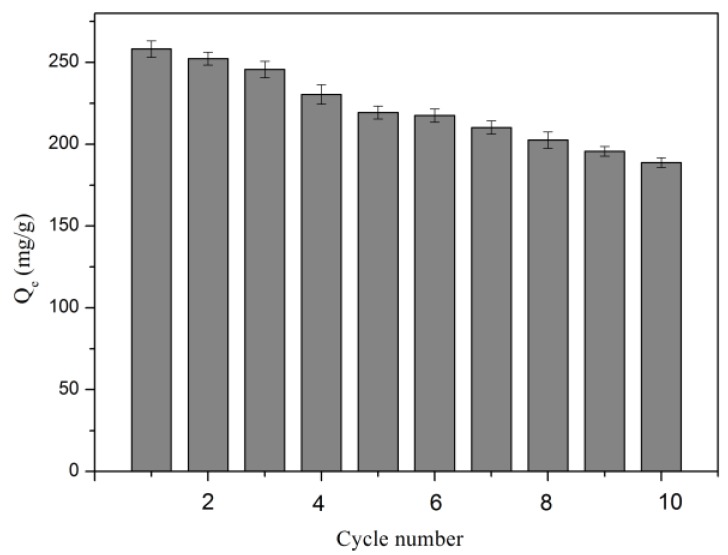
Pb(II) adsorption capacity of CLNPs at different regeneration cycles (CLNP dosage = 15 mg/50 mL, *C_0_* = 100 mg/L, temperature = 30 °C, pH = 6.03).

**Table 1 molecules-24-02704-t001:** The contents of functional groups in AL and CLNPs.

Lignin	Aliphatic OH (150–145.7 ppm)	Condensed Phenolic OH (145–140.7 ppm)	Guaiacyl and Catechol OH (140–137.6 ppm)	Total Phenolic OH	Carboxyl (136–133.8 ppm)
	(mmol/g)	(mmol/g)	(mmol/g)	(mmol/g)	(mmol/g)
AL	2.32	1.76	1.42	3.18	0.95
CLNPs	3.02	1.60	0.43	2.03	1.80

**Table 2 molecules-24-02704-t002:** Comparison of the lead adsorption capacity by CLNPs and other adsorbents.

Adsorbent	T (°C)	Time (min)	pH	K_2_	Q_e_ (mg/g)	Reference
Zeolite A	25	30	7.5	-	213.0	[28]
Commercial active carbon	25	360	6.0	0.005	29.2	[29]
Pb–ITMCB	40	480	6.0	2.9 × 10^−4^	259.7	[30]
Lignosulfonate sphere	30	150	5.0	0.02	27.1	[17]
Lignin-grafted carbon nanotubes	25	60	5.8	0.03	235.0	[31]
Carboxymethylation formic lignin	30	-	6.0	-	107.5	[16]
CLNPs	30	180	6.03	0.0544	333.26	This study

**Table 3 molecules-24-02704-t003:** Kinetic parameters for Pb(II) adsorption onto CLNPs.

Sample	Pseudo-First-Order Kinetic			Pseudo-Second-Order Kinetic		
	Qe (mg/g)	K_1_ (1 min^−1^)	R^2^	Qe (mg/g)	K_2_ (g/mg min)	R^2^
CLNPs	304.0	1.08	0.6013	350.9	0.0544	0.9991

**Table 4 molecules-24-02704-t004:** Freundlich and Langmuir isotherm model parameters for the adsorption of Pb(II) onto CLNPs.

Sample	Langmuir Model			Freundlich Model		
	B (L/mg)	Qmax (mg/g)	R^2^	n	K_F_ (mg/g)	R^2^
CLNPs	0.1734	312.5	0.973	2.86	63.09	0.448

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
