# Peer review of "Preparation of a Novel Lignin Nanosphere Adsorbent for Enhancing Adsorption of Lead"

_molecules, 2019, doi:10.3390/molecules24152704_

Reviewer 1 Report

This study investigated “Preparation of a novel lignin nanosphere adsorbent for enhancing adsorption of lead”.  Some of issues should be addressed.

1.      The authors have to elaborate the significance of the manuscript. There are plenty of works in this subject regarding low cost adsorbent.

2.      Page 2, line 67, content of Ionic liquids[EMIM] should be addressed.

3.      Page 2, line 74-75, microwave operation conditions should be listed in detail.

4.      The authors should compare their findings with other people’s works in literature.

5.      The separation process of CLNPs after adsorption might have to be addressed as well. (for future applications or reuse)

6.      The potential drawbacks or limitation should be addressed as well.

Author Response

Great thanks for your suggestion. We have also invited several researchers to revise the English usage in the manuscript. And some revisions labeled in RED have been made in the part of introduction.

Comments and Suggestions for Authors

This study investigated “Preparation of a novel lignin nanosphere adsorbent for enhancing adsorption of lead”.  Some of issues should be addressed.

1. The authors have to elaborate the significance of the manuscript. There are plenty of works in this subject regarding low cost adsorbent.

Thanks very much for your hard-working and kind suggestions. Although plenty of works have been published focus on low cost adsorbent, as expert said, no report on carboxyl modified lignin nanospheres has been reported to date to develop the application of bioresources. The prepared carboxymethyl lignin nanospheres in this study have a larger adsorption capacity for heavy metal lead than the relevant literature with excellent reusability for metal ions adsorption, and the solvents of ILs can be recycled also, which made it great potential for industrial applications in wastewater treatments.

As suggested, some revisions labeled in RED have been made.

2. Page 2, line 67, content of Ionic liquids [EMIM] should be addressed.

As suggested, some revisions labeled in RED have been made.

3. Page 2, line 74-75, microwave operation conditions should be listed in detail.

Great thanks for your suggestion. Some revisions labeled in RED have been made.

4. The authors should compare their findings with other people’s works in literature.

Great thanks for your suggestion. Some revisions labeled in RED have been made.

5. The separation process of CLNPs after adsorption might have to be addressed as well. (for future applications or reuse)

As suggested, the separation process of CLNPs after adsorption has been added to the corresponding positions in the article and some revisions labeled in RED have been made.

6. The potential drawbacks or limitation should be addressed as well.

Great thanks for your hard-working, some revisions labeled in RED have been made.

Reviewer 2 Report

The manuscript entitled "Preparation of a novel lignin nanosphere adsorbent for enhancing adsorption of lead" prepared by Chao Liu, Youming Li and Yi Hou, concerns novel lead adsorber based on the lignin nanosphere. The Authors present reliable and imposing results on the achieved lead adsorber, which can compete with the currently known systems dedicated to remove lead from the wasted water. The Authors provided reliable literature search for both, the Introduction as well as Experimental Section, and inform the Reader about the current challenges in the field, but also limitations, which come from the utilized approaches and materials. However, the Authors did not avoid some errors/mistakes (mainly editorial ones), and their number is too high in order to read the article without doubts.

In my opinion, very few minor issues should be considered to make this manuscript suitable for publication in the Molecules journal. They are listed below:

1) There is plenty of editorial mistakes, which successfully complicate reading of the manuscript. There are many places, where dot sign "." is repeated without reason, then not all of the equations are numbered (Langmuir or  Freundlich ones, lines 220 and 221), then insets are not readable if consider axis labels (Figure 8 and 9). Another mistake is in the line 104 - should be "M" (mol/L), not "m". Please, pay more attention to provide good quality manuscript in order to be easy to follow by the future Readers.

2) In Figure 12. you have shown the regeneration cycles for CLNPs lead adsorber. You stated that after 5 cycles (by the way, in the line 21 should be "five", no "fives") the product efficiency decreases on about 15%. Could you provide more data about reusable of the lead adsorber? I mean, it could be valuable to see the efficiency after 10 cycles, however with the higher number of the performed cycles, like 100 or 200, reliable data could be provided by proper calculations in order to stress out more the high efficiency of the achieved lead adsorber. I recommend to perform experimentally 10 full cycles - add the new data to the Figure 12., and then calculate the value of losses for 100 cycles, or contrary - define the number of cycles, when your adsorber will characterize just 50% of efficiency.

Independently of the aforementioned comments, I would like to appreciate performed experiments and achieved results.

Author Response

As suggested, some revisions labeled in RED have been made.

Comments and Suggestions for Authors

The manuscript entitled "Preparation of a novel lignin nanosphere adsorbent for enhancing adsorption of lead" prepared by Chao Liu, Youming Li and Yi Hou, concerns novel lead adsorber based on the lignin nanosphere. The Authors present reliable and imposing results on the achieved lead adsorber, which can compete with the currently known systems dedicated to remove lead from the wasted water. The Authors provided reliable literature search for both, the Introduction as well as Experimental Section, and inform the Reader about the current challenges in the field, but also limitations, which come from the utilized approaches and materials. However, the Authors did not avoid some errors/mistakes (mainly editorial ones), and their number is too high in order to read the article without doubts.

Thanks very much for your positive regards for the work. As your kind suggestions some revisions labeled in RED have been made. We have also invited several researchers to revise the English usage in the manuscript. And we hope that all the treatment could be acceptable.

In my opinion, very few minor issues should be considered to make this manuscript suitable for publication in the Molecules journal. They are listed below:

1)     There is plenty of editorial mistakes, which successfully complicate reading of the manuscript. There are many places, where dot sign "." is repeated without reason, then not all of the equations are numbered (Langmuir or Freundlich ones, lines 220 and 221), then insets are not readable if consider axis labels (Figure 8 and 9). Another mistake is in the line 104 - should be "M" (mol/L), not "m". Please, pay more attention to provide good quality manuscript in order to be easy to follow by the future Readers. 

Great thanks for your suggestion. I'm sorry some clerical errors should not appearand as suggested, some revisions labeled in RED have been made.              

2) In Figure 12. you have shown the regeneration cycles for CLNPs lead adsorber. You stated that after 5 cycles (by the way, in the line 21 should be "five", no "fives") the product efficiency decreases on about 15%. Could you provide more data about reusable of the lead adsorber? I mean, it could be valuable to see the efficiency after 10 cycles, however with the higher number of the performed cycles, like 100 or 200, reliable data could be provided by proper calculations in order to stress out more the high efficiency of the achieved lead adsorber. I recommend to perform experimentally 10 full cycles - add the new data to the Figure 12., and then calculate the value of losses for 100 cycles, or contrary - define the number of cycles, when your adsorber will characterize just 50% of efficiency.

Independently of the aforementioned comments, I would like to appreciate performed experiments and achieved results.

Great thanks for your suggestion. In fact, we did more than ten sets of re-use tests, and the relevant experimental data have been added to the corresponding positions in the article. We hope that all the treatment could be acceptable. Thank you again.

Round  2

Reviewer 1 Report

The revision of the manuscript is satisfactory.